# Role of the Hox Genes, *Sex combs reduced*, *Fushi tarazu* and *Antennapedia*, in Leg Development of the Spider Mite *Tetranychus urticae*

**DOI:** 10.3390/ijms241210391

**Published:** 2023-06-20

**Authors:** Xiang Luo, Yu-Qi Xu, Dao-Chao Jin, Jian-Jun Guo, Tian-Ci Yi

**Affiliations:** 1Guizhou Provincial Key Laboratory for Agricultural Pest Management of Mountainous Regions, Institute of Entomology, Guizhou University, Guiyang 550025, China; lx1205340985@163.com (X.L.); 15286708491@163.com (Y.-Q.X.); dcjin@gzu.edu.cn (D.-C.J.); 2Scientific Observing and Experimental Station of Crop Pests in Guiyang, Ministry of Agriculture and Rural Affairs of the People’s Republic of China, Guiyang 550025, China

**Keywords:** *Tetranychus urticae*, Hox genes, appendage development, ontogeny

## Abstract

Mites, the second largest arthropod group, exhibit rich phenotypic diversity in the development of appendages (legs). For example, the fourth pair of legs (L4) does not form until the second postembryonic developmental stage, namely the protonymph stage. These leg developmental diversities drive body plan diversity in mites. However, little is known about the mechanisms of leg development in mites. Hox genes, homeotic genes, can regulate the development of appendages in arthropods. Three Hox genes, *Sex combs reduced* (*Scr*), *Fushi tarazu* (*Ftz*) and *Antennapedia* (*Antp*), have previously been shown to be expressed in the leg segments of mites. Here, the quantitative real-time reverse transcription PCR shows that three Hox genes are significantly increased in the first molt stage. RNA interference results in a set of abnormalities, including L3 curl and L4 loss. These results suggest that these Hox genes are required for normal leg development. Furthermore, the loss of single Hox genes results in downregulating the expression of the appendage marker *Distal-less* (*Dll*), suggesting that the three Hox genes can work together with *Dll* to maintain leg development in *Tetranychus urticae*. This study will be essential to understanding the diversity of leg development in mites and changes in Hox gene function.

## 1. Introduction

Appendages, as the tool for arthropod survival, exhibit a rich diversity in morphology and function, which in turn has driven the evolution of the arthropod body plan. There iss a special group of arthropods, the mites, which usually only have one pair of chelicerae and four pairs of walking legs. Their legs, unlike those of insects, do not have an abundance of functions but exhibit a specific developmental pattern in leg development. For example, the fourth pair of legs (L4 in short) does not develop when the mite larvae hatch, i.e., the larvae have three pairs of legs. Instead, this occurs after the larvae have molted into protonymph, the second postembryonic developmental stage. However, in some special groups (such as Erophyidae and Podapolipidae), mites only have two or three pairs of walking legs. So, we should ask what genetic factors regulate the leg development of mites and whether the the developmental pathways are the same for different legs?

Hox genes, homeotic genes, were the first genes found to specify the identity of the arthropod appendage [1,2,3]. For example, expression alterations in the *Sex combs reduced* (*Scr*) gene resulted in the labium transitioning to a state of a mixed identity [4]. This discovery led to the idea that changes in the regulation and function of Hox genes could drive the evolution of the arthropod appendage developmental diversity [5,6]. Hox genes are expressed in different segments along the anterior–posterior axis of the embryo, encode a set of homologous domain transcription factors, and determine the identity of the appendages in the corresponding segments [7]. For example, in insects, Hox genes were able to determine the formation of appendages in different parts of the body. *Antennapedia* (*Antp*) gene is expressed in the thorax where walking legs develop [8,9,10], whereas *Ultrabithorax* (*Ubx*) gene inhibits leg formation in the abdomen [11]. The interaction of Hox proteins with their associated upstream and downstream factors is known as the Hox gene pathway, and these factors contain proteins that bind DNA in concert with Hox proteins and targeted cis-regulatory elements [12]. Changes in this pathway are associated with the formation of appendages; for example, changes in *Antp* expression were associated with the development of thoracic legs in the *Bombyx mori* [13] and the inhibition of abdominal leg formation in water flea *Daphnia* [14]. Differences in the target genes of Hox genes also affect appendage development; in spiders, *Antp* inhibited the formation of the abdominal legs, while its misexpression in *Drosophila melanogaster* resulted in the conversion of arista to tarsus [15]. The changes in Hox protein can lead to failure of appendage formation. For example, the formation of a novel structural domain in the *Ubx* protein of insects resulted in the inhibition of ventral limb formation [16,17]. These examples show that Hox genes play a crucial role in the development of appendages in arthropods.

Previous studies on Hox gene expression in mites showed that *Scr*, *Ftz*, and *Antp* genes were expressed in leg segments [18,19]. Thus, these genes may play a role in leg development in mites. However, the function of these genes in mites has not been reported. In this study, we cloned and identified *Scr*, *Ftz* and *Antp* genes of *Tetranychus urticae*, a potential genetic study species in mites [20]. We investigated the function of *Scr*, *Ftz*, and *Antp* genes, using RNA interference (RNAi) to clarify the effect of Hox genes in leg development. Furthermore, we analyzed the expression of *Distal-less* (*Dll*, a limb-promoting gene) in response to Hox genes RNAi samples to illuminate the cooperative relationship between these genes and *Dll* in leg development. This study will provide an important contribution to our understanding of the diversity of mite leg development and changes in Hox gene function.

## 2. Results

### 2.1. Hox Genes Identification, Multiple Sequence Alignment Analysis and Expression Patterns in Different Developmental Stages

Using the BLAST search function in NCBI, *TuScr*, *TuFtz*, and *TuAntp* genes were discovered. The full length of these genes was obtained by PCR from adult cDNA. These nucleotide sequences of *TuScr*, *TuFtz* and *TuAntp,* respectively, were 1300, 1226 and 1363 bp, and the open reading frames (ORFs) were predicted to encode 349, 366 and 375 amino acid residues with a highly conserved YPWM motif and homeodomain (HD). When these deduced amino acid sequences were aligned with other known *Scr*, *Ftz*, and *Antp* genes, we found that YPWM motif and HD were highly conserved across all the species (Figure 1). All sequence data were submitted to the GenBank database under the accession numbers OQ079716, OQ079717 and OQ079718. Relative expressions of these Hox genes were measured by quantitative real-time PCR (RT-qPCR) after sampling each state of *T. urticae* (Figure 2). The results showed that expression levels of three Hox genes had the highest expression levels in the first molting state and implied that these genes might be involved in L4 formation during the molting.

### 2.2. RNA Interference of the Individual Hox Genes Results in L4 Loss

To investigate the role of Hox genes in development of legs, topical RNAi was used to silence *TuScr*, *TuFtz* or *TuAntp*. We dripped ds*GFP* onto control larvae following the same scheme. RNAi efficiencies of Hox genes in the protonymph stage were examined using RT-qPCR. The results indicated that the levels of *TuScr*, *TuFtz* and *TuAntp* transcripts were significantly reduced in their respective knockdown conditions (ds*TuScr*, ds*TuFtz* and ds*TuAntp*) with 78.39%, 95.31% and 71.78% reductions, respectively, (*p* = 0.0018, *p* = 0.0018 and *p* = 0.0270) (Figure 3A) compared to the ds*GFP*-control, which suggested that the silencing effect of topical RNAi was effective. In the phenotype, no morphological abnormalities were observed in control samples treated with ds*GFP*. Wild-type (WT) protonymphs formed L4 normally, and other legs had no abnormal morphology (Figure 3D). However, the respective knockdowns of *TuScr*, *TuFtz* and *TuAntp* led to L4 development failure (Phenotypic rate: 11.67% 27.89% and 10.91%) (Figure 3C), and mites with L4 loss could not successfully molt the old cuticles from their dorsa (Figure 3D). In the ds*TuFtz*–treated group, a few mites exhibited the phenotype of L3 curl (1.92%). Mites with L3 curl had reduced mobility, while mites with L4 loss died due to an inability to move, compared to ds*GFP*-treated varieties. Furthermore, to determine the roles of *TuScr*, *TuFtz* and *TuAntp* in regulating *TuDll* expression, we examined *TuDll* expression in treated samples using RT-qPCR. After the respective treatments, the relative expression levels of *TuDll* were downregulated in ds*TuAntp*-treated samples, albeit not significantly (Figure 3B).

### 2.3. RNA Interference of Combined Hox Gene Results in L3 Curl

The respective treatments of ds*TuScr*, ds*TuFtz* and ds*TuAntp* resulted in L4 loss of protonymphs, but the phenotype of L3 curl was not shown in ds*TuScr* or ds*TuAntp*. Therefore, we designed dual and triple RNAi of Hox gene combinations, i.e., *TuScr* and *TuFtz* (ds*TuScr*/*Ftz*), *TuScr* and *TuAntp* (ds*TuScr*/*Antp*), *TuFtz* and *TuAntp* (ds*TuFtz*/*Antp*) and *TuScr*, *TuFtz* and *TuAntp* (ds*TuScr*/*Ftz*/*Antp*). These knockdown conditions were selected to verify whether an even more pronounced effect could be generated.

RT-qPCR analysis of Hox genes transcript levels in protonymph revealed that *TuScr*, *TuFtz* and *TuAntp* were significantly downregulated in all knockdown conditions (Figure 4A). In terms of phenotype, all combination conditions resulted in L3 curl (Phenotypic rate: ds*TuScr*/*Ftz*/*Antp* 8.51%; ds*TuScr*/*Ftz* 22.81%; ds*TuScr*/*Antp* 1.81%; dsTuFtz/Antp 1.83%) while, except for ds*TuScr*-/*Ftz*-/*Antp*-treated examples, multiple abnormal phenotypes were found in combination conditions. In detail, only L3 curl was found in ds*TuScr*-/*Ftz*-/*Antp*-treated samples, while L3 curl or L4 loss were found in conditions stemming from other combinations. In addition, the use of ds*TuScr*-/*Ftz*-treated samples resulted in L3 curl and L4 tissue loss (1.92%), i.e., L4 with only the outer epidermis and loss of inner tissue, and ds*TuScr*-/*Antp*-treated resulted in L4 curl (0.93%) (Figure 4C).

## 3. Discussion

In this paper, we identified three Hox genes and studied their functions in relation to leg development. These genes contained a homeodomain identical to that in other arthropods (Figure 1). The temporal analyses indicated that *TuScr*, *TuFtz* and *TuAntp* had the potential to play roles in leg development. Further functional studies showed that both L3 and L4 developments were affected by these genes, and that the regulatory pathways of these Hox genes in L3 and L4 were different. This was because the loss of a single gene affected the development of L4, while L3 required the loss of multiple genes to be affected. These results indicated that the three Hox genes play important roles in the leg development of mites.

In most arthropods, the function of the Hox genes is the homeotic function, and the loss or abnormality of these genes leads to changes in the identity of the appendage [21,22,23,24]. In addition, recent studies have shown that the function of some Hox genes has changed. For example, the role of *Antp* in spiders was found to inhibit leg development [15], while the role of *Ftz* in *D. melanogaster* was not assessed as being related to appendage development [25]. Our data suggested that single or combined RNAi with *TuFtz TuScr* and *TuAntp* caused leg dysplasia rather than altering leg identity. This role is reminiscent of the loss of *Antp* in silkworms, which leads to defects in thoracic leg development rather than a homeotic transformation [13,26]. Our results can be explained by the fact that Hox genes, *TuScr*, *TuFtz*, and *TuAntp*, act as leg promoters that act together to maintain L3 and L4 development. Interestingly, the three Hox genes appeared to be different in the regulatory pathways of L3 and L4 because L3 required the combined action of multiple Hox genes, which only affected leg development, whereas both or three Hox genes were impaired. However, in L4, the loss of any one of these genes affected leg development. Therefore, we hypothesized that *TuScr*, *TuFtz*, and *TuAntp’s* regulatory pathways differed in L3 and L4.

In summary, unlike most arthropods, the role of Hox genes in mites was to promote leg development. Previous studies have shown that this functional change may be related to changes in the Hox proteins during evolution [27,28,29,30,31]. For example, specific motifs (QA domain) evolved in the C-terminus of *Ubx* protein [16,17], new protein regions evolved in the *Antp* N-terminus [32] and changes emerged in the YPWM motif of *Ftz* proteins [25], all of which resulted in a loss of homeotic function. However, the motifs of the Hox proteins in *T. urticae* were not altered (Figure 1). Therefore, we hypothesize that the altered function of the Hox genes in *T. urticae* may not be related to the modified protein motifs.

On the other hand, cofactors or target genes of Hox genes may also lead to functional alterations [7], as one Hox gene may involve hundreds of cofactors [33,34]. It is also not unheard of for different cofactors to cause alterations in Hox gene function. For example, the role of *Antp* in spiders was to inhibit leg development, but misexpression in *Drosophila* led to arista transformation to having a tarsal identity [15]. Therefore, we speculate that different or altered cofactors in *T. urticae* led to changes in Hox gene function, as previous studies have shown that mite genomes are much smaller than those of other arthropods [35] and that Hox genes deletion varies among different mites [36,37,38]. In this regard, it has been suggested that *Abdominal-A* (*Abd-A*) was responsible for the morphological diversity of mites [39]. In addition, the relationship between Hox genes and the target gene (*DlI*) differed significantly in different arthropods. For example, in the abdomen of the more evolved pterygota of insects, Hox genes repressed the expression of *Dll* [11,40,41], which in turn inhibited the formation of the abdominal legs. Conversely, in the more primitive microcoryphia, Hox genes were non-repressive to *Dll* [42]. This suggests that the relationship between the action of Hox genes on the same target genes was variable in different arthropods. The loss of the *Dll* in *T. urticae* failed in leg development [20], which was consistent with our results which showed that that in *T. urticae,* Hox genes and *Dll* act together in leg development. These results propose that alterations of Hox gene function in *T. urticae* may be caused by cofactors or target genes. Based on this result, our future work will focus on regulating the Hox genes in *T. urticae* and their target genes and cofactors.

## 4. Materials and Methods

### 4.1. Mites Culture

The *T. urticae* has been routinely reared on the leaf of a kidney bean, under conditions of controlled temperature (27 ± 1 °C), photoperiod (L:D, 14 h:10 h) and relative humidity (65 ± 5%) in the laboratory of the Institute of Entomology, Guizhou University. In order to obtain certain unique stages of mites for experimental treatment, leaf discs were made as described in [43]. Based on this experimental setup, we collected different developmental stages of *T. urticae* (eggs, larvae, protonymph, deutonymph, adults and states in three molts) to measure the expression of Hox genes.

### 4.2. Molecular Cloning and Multiple Sequence Alignment of Genes

Total RNA was extracted using the MiniBEST Universal RNA Extraction Kit (Takara Biomedical Technology, Beijing, China) according to the instructions in the manual. First-strand cDNA was synthesized using Starscript II (Gene Star, Beijing, China) and RNA and cDNA concentrations were measured by a NanoDrop 2000c spectrophotometer (Thermo Scientific, Waltham, MA, USA). The quality samples were stored at −80 °C and −20 °C for use in subsequent experiments. All primers were designed by Primer v.6.0 software (Appendix A). The mite Hox genes cloning was performed as described in [44]. In addition, the PCR products amplified by each primer were also sequenced to confirm their specificity.

Multiple sequence alignment was performed based on the protein sequences of *TuScr*, *TuFtz* and *TuAntp* to determine the identity of these genes. In order to demonstrate the conserved nature of these genes, we used *Drosophila melanogaster*, *Tribolium castaneum* and *Achaearanea tepidariorum*. All these sequences were obtained from NCBI (http://www.ncbi.nlm.nih.gov/) (accessed on 17 April 2021) and their GenBank accession numbers, respectively, are AAA19240.1, AAK16422.1, NP477498.1, AAK16421.1, AAA28373.1, EEZ99250.1 and HE608680.1. All the conserved sequences were aligned with the CLUSTALW online program.

### 4.3. Preparation of the dsRNA Formulation and the RNAi Effect Detection

The dsRNA synthesis was done using the Transcript T7 High Yield Transcription Kit (Thermo Scientific, Shanghai, China) according to the instructions in the manual. After synthesis, dsRNA was purified using the Gene JET RNA Purification Kit (Thermo Scientific). Our topical method referred to previously existing topical methods [44,45]. The topical method was applied to deliver dsRNA with SYS-PV830 (WPI Company, Sarasota, FL, USA). The microelectrode needle puller PUL-1000 (WPI Company) was used to make TW100-4 (WPI Company) at the conditions of 50 (heat), 50 (force), 8 (distance) and 0 (delay). To verify whether *TuScr*, *TuFtz* and *TuAntp* genes affect the leg development of mites during molting, 50 larvae (before molting 2 h) were placed on each leaf-disk. The dsRNA solution was placed on the dorsa of the larvae. Approximately 5–10 nL of dsRNA (8 μg/μL) was placed on each mite. After 24 h, the live mites of topical RNAi were collected per replicate for the extraction of total RNA. In addition, ds*GFP*-treated larvae underwent the same operation in the same conditions as the controls. The treatment and control groups were recorded after 24 h for use in phenotypic analysis.

### 4.4. Real-Time Quantitative PCR

The RT-qPCR was performed on a StepOnePlus Real-Time PCR System (Applied Biosystems, Foster City, CA, USA). The 10.0 μL reaction system contained 5.0 μL 2x RealStar Green Fast Mixture (GeneStar, Bieijing, China), 0.5 μL forward and 0.5 μL reverse primers, 0.5 μL cDNA and 3.5 μL RNase-free water. We analyzed the expression pattern of target genes and produced all graphs by GraphPad Prism v.8.0 software. The relative expression levels were calculated using the method of 2^−ΔΔCt^. Experimental data were analyzed by performing tests in IBM-SPSS v.21.0 (IBM, Armonk, NY, USA) to compare the differences between the treatment and control groups. The house-keeping gene *ATP* was used to normalize gene expression levels [46].

## 5. Conclusions

We identified three key genes involved in leg development in *T. urticae*. Their expression pattern indicated that the highest expression was reached before leg formation, suggesting their involvement in regulating this important process in leg development. In addition, this view was confirmed by the silencing of Hox genes *TuScr*, *TuFtz* and *TuAntp*, which caused abnormal leg development and repressed the expression of the limb-promoting gene (*Dll*), suggesting their involvement in leg development in *T. urticae*. In summary, our study provided a role for Hox genes in spider mites, and these results provided evidence for Hox genes-mediated leg diversification.

## Figures and Tables

**Figure 1 ijms-24-10391-f001:**
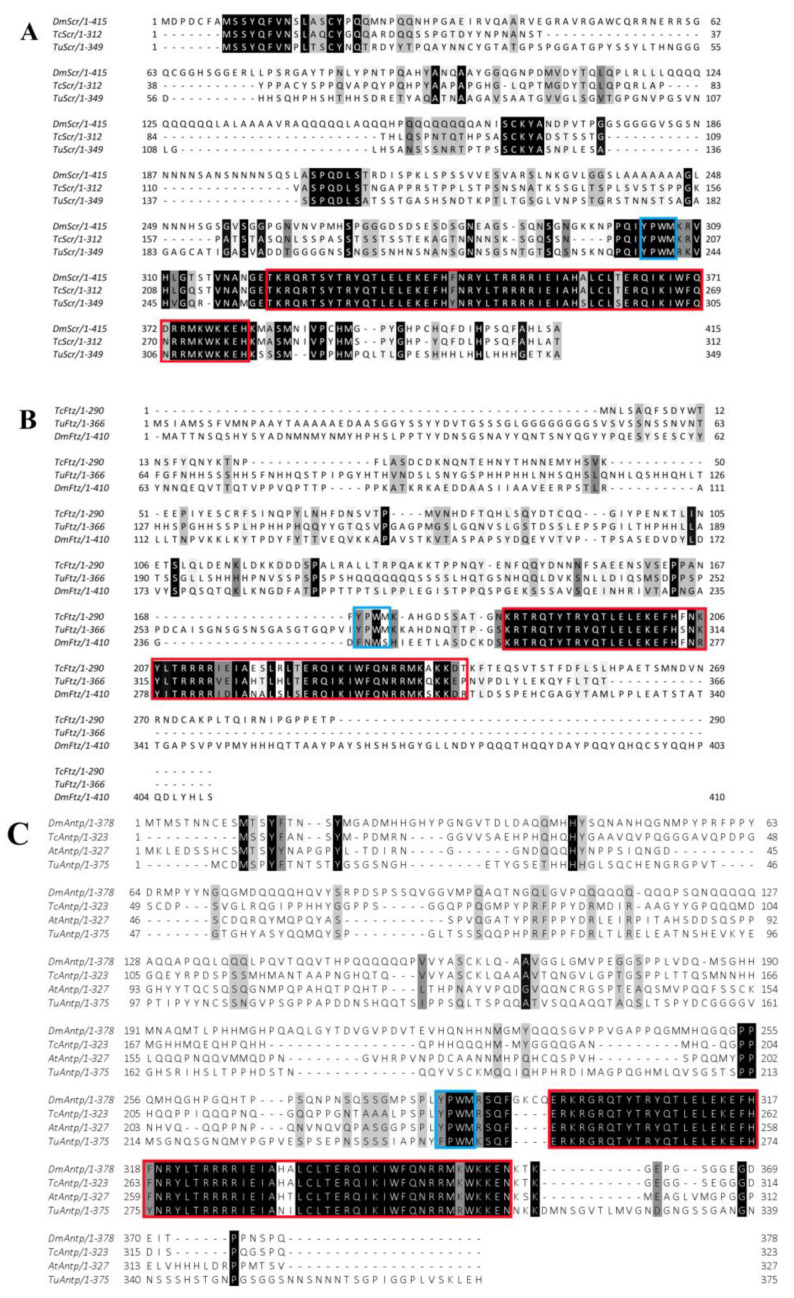
Domain structure and multiple sequence alignment of *TuScr*, *TuFtz* and *TuAntp*. (**A**) *TuFtz*; (**B**) *TuScr*; (**C**) *TuAntp*. The blue box is the YPWM motif, and the red box is the Homebox domain. Amino acid sequence alignment of *TuScr*, *TuFtz* and *TuAntp* orthologs from *Drosophila melanogaster*, *Tribolium castaneum* and *Achaearanea tepidariorum*.

**Figure 2 ijms-24-10391-f002:**
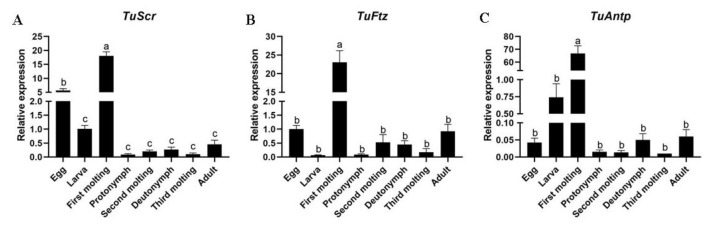
Mean (±SEM) relative expression of three Hox genes in eight developmental stages of *Tetranychus urticae*. (**A**). *TuScr* multi Sequence alignment. (**B**). *TuFtz* multi Sequence alignment. (**C**). *TuAntp* multi Sequence alignment. Means capped with different letters are significantly different (Tukey’s HSD test: *p* < 0.05).

**Figure 3 ijms-24-10391-f003:**
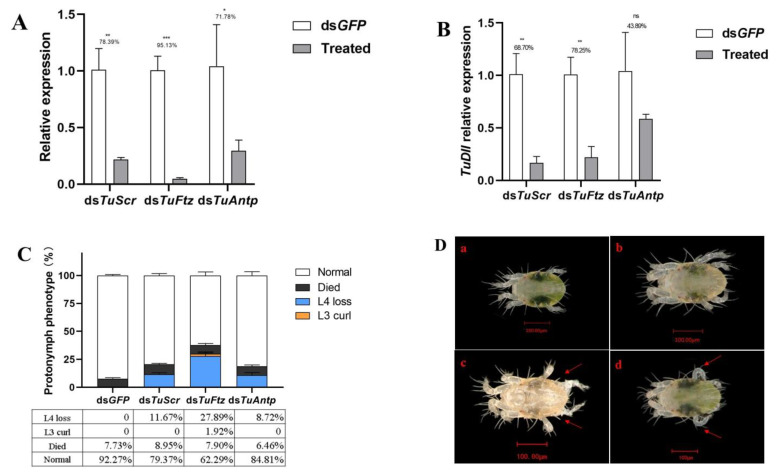
Effects of RNAi-mediated knockdown of the individual Hox genes (*TuScr*, *TuFtz* or *TuAntp*) in leg development of *T. urticae*. (**A**) Relative expressions of Hox genes were detected by RT-qPCR when the mites were treated with ds*TuScr*, ds*TuFtz* or ds*TuAntp* at 24 h. (**B**) Relative expressions of *TuDll* in protonymph after gene silencing of *TuScr*, *TuFtz* and *TuAntp*. (**C**) Phenotypic rate caused by single RNAi with Hox genes. (**D**) Abnormal phenotype caused by RNAi with Hox genes (a. larvae; b. wild-type protonymph after ds*GFP*; c. protonymph with L4 loss; d. protonymph with L3 curl). Abnormal phenotype marked with arrows. The significant difference between the two groups was indicated with ‘*’ (‘*’, *p* < 0.05. ‘**’, *p* < 0.01. ‘***’, *p* < 0.001. ‘ns’ presented no significant difference, Student’s *t* test).

**Figure 4 ijms-24-10391-f004:**
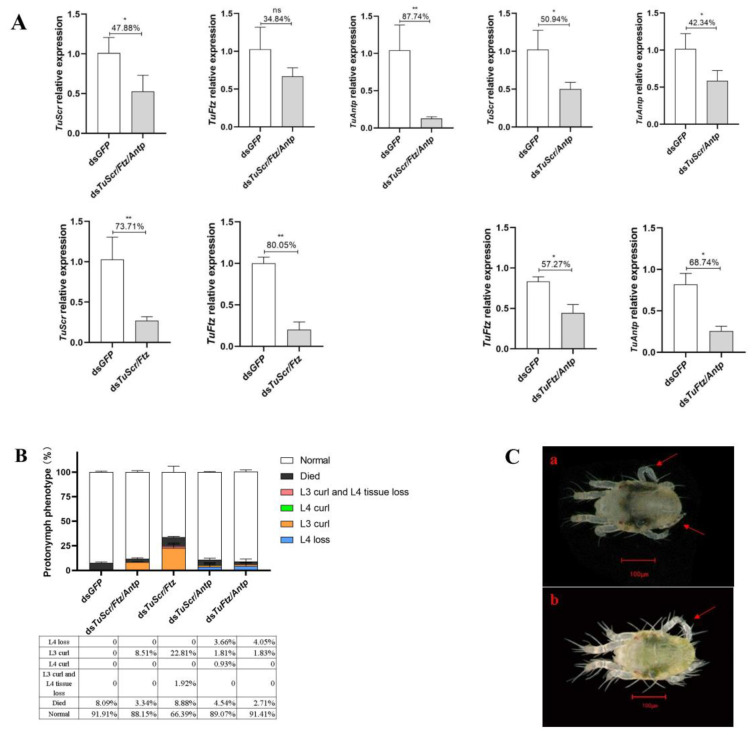
Effects of RNAi-mediated knockdown of the combined Hox genes (*TuScr*, *TuFtz* or *TuAntp*) in leg development of *T. urticae*. (**A**) Relative expressions of *TuScr*, *TuFtz* or *TuAntp* in different combinations of RNAi. (**B**) Phenotypic rate caused by combined RNAi with Hox genes. (**C**) Abnormal phenotype caused by combined RNAi with Hox genes (a. protonymph with L3 curl and L4 tissue loss; b. protonymph with L4 curl). Abnormal phenotype marked with arrows. The significant difference between the two groups was indicated with ‘*’ (‘*’, *p* < 0.05. ‘**’, *p* < 0.01. ‘ns’ was presented no significant difference, Student’s *t* test).

## Data Availability

Data is contained within the article.

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
