# Peer review of "Role of the Hox Genes, *Sex combs reduced*, *Fushi tarazu* and *Antennapedia*, in Leg Development of the Spider Mite *Tetranychus urticae"

_ijms, 2023, doi:10.3390/ijms241210391_

Round 1

Reviewer 1 Report

In this paper: "Role of the Hox genes, Sex combs reduced, Fushi tarazu and An-2 tennapedia, in leg development of the spider mite Tetranychus 3 urticae"  the authors  analyzed and explained the diversity of leg development in mites and changes in Hox gene function. After implementing the discussion, paper can be published.

More detailed comments by reviewer

This paper provides a contribution to the knowledge of the diversity leg development in mites according to the  Hox gene function. The paper was conducted with a careful and clear methodology and contributes to the knowledge on the function of genes in the morphological development of mites.

The methodology used for the analyzes is explained scrupulously and carefully. To a scientific point of view the work is good for publication, after having increased the paragraph of the conclusions.

Reviewer 2 Report

The main question addressed by this research is the following: This study will be essential to understanding the diversity of leg development in mites and changes in Hox gene function. I am not sure if the topic can be relevant in the field. I am not sure if it add to the subject area, compared with other published material.

I think the article should highlight what are the practical consequences of its conclusions, and to what extent the mite population could be controlled, if possible.

The conclusions are consistent with the evidence presented, but the ultimate application of these studies is not at all clear, which should be explained. The bibliographical references are not current and do not cover a wide range of articles that support the research developed. Tables and Figures contain the correct scientific information and with the necessary statistical parameters to give them credibility.

Overall rating

The article does not deserve to be published because it does not meet all the necessary standards to be a reference.

The English is O.K.

Round 2

Reviewer 2 Report

This study might be essential to understanding the diversity of leg development in mites and changes in Hox gene function.

The article has notably improved in terms of comprehension, due to the modifications carried out by the authors. The figures provide convincing information, and furthermore, the statistical parameters are correct. The bibliography is up to date and the work detailed here could be an interesting point of support for more specific studies in the near future.

This study might be essential to understanding the diversity of leg development in mites and changes in Hox gene function.

The article has notably improved in terms of comprehension, due to the modifications carried out by the authors. The figures provide convincing information, and furthermore, the statistical parameters are correct. The bibliography is up to date and the work detailed here could be an interesting point of support for more specific studies in the near future.
